# 14-3-3-protein regulates Nedd4-2 by modulating interactions between HECT and WW domains

Pavel Pohl[1,2], Rohit Joshi[1,3], Olivia Petrvalska[1,3], Tomas Obsil [1,3✉] & Veronika Obsilova [1✉]

Neural precursor cell expressed developmentally down-regulated 4 ligase (Nedd4-2) is an E3 ubiquitin ligase that targets proteins for ubiquitination and endocytosis, thereby regulating numerous ion channels, membrane receptors and tumor suppressors. Nedd4-2 activity is regulated by autoinhibition, calcium binding, oxidative stress, substrate binding, phosphorylation and 14-3-3 protein binding. However, the structural basis of 14-3-3-mediated Nedd4-2 regulation remains poorly understood. Here, we combined several techniques of integrative structural biology to characterize Nedd4-2 and its complex with 14-3-3. We demonstrate that phosphorylated Ser$^{342}$ and Ser$^{448}$ are the key residues that facilitate 14-3-3 protein binding to Nedd4-2 and that 14-3-3 protein binding induces a structural rearrangement of Nedd4-2 by inhibiting interactions between its structured domains. Overall, our findings provide the structural glimpse into the 14-3-3-mediated Nedd4-2 regulation and highlight the potential of the Nedd4-2:14-3-3 complex as a pharmacological target for Nedd4-2-associated diseases such as hypertension, epilepsy, kidney disease and cancer.

[1] Department of Structural Biology of Signaling Proteins, Division BIOCEV, Institute of Physiology of the Czech Academy of Sciences, Vestec, Czech Republic. [2] 2nd Faculty of Medicine, Charles University, Prague, Czech Republic. [3] Department of Physical and Macromolecular Chemistry, Faculty of Science, Charles University, Prague, Czech Republic. ✉email: obsil@natur.cuni.cz; veronika.obsilova@fgu.cas.cz

The neural precursor cell expressed developmentally down-regulated 4 (Nedd4-2) is a member of the HECT E3 ubiquitin ligase family. As such, this enzyme targets proteins for ubiquitination in mammalian programmed cell death[1,2]. Mouse knockout studies have confirmed that Nedd4-2 plays a key role in animal physiology by regulating multiple substrates, including the epithelial sodium channel (ENaC). High ENaC activity and blood pressure with aberrant renal $Na^+$ reabsorption are observed in Liddle syndrome resulting from mutations in its Nedd4-2-interacting motif[3,4]. In addition to regulating ion transport, Nedd4-2 controls cellular trafficking in different tissues, modulating multiple signaling pathways through these interactions. Unsurprisingly, respiratory distress, hypertension, and electrolyte imbalance and kidney disease stand out among the pathological consequences of Nedd4-2 dysregulation, in line with mouse studies associating numerous SNPs in the Nedd4-2 gene with these conditions and with multiple tumor types[5,6]. Therefore, potential therapeutic interventions may be developed by targeting the ubiquitin system for drug development via Nedd4-2 interactions[7].

Developing such strategies targeting Nedd4-2 requires thoroughly understanding the structural-functional relationships of this protein. Fortunately, all nine members of the Nedd4 family of mammalian HECT E3 ligases have a similar modular multi-domain architecture, typically consisting of an N-terminal C2 domain, two-to-four WW domains, which contain two conserved tryptophan residues and a proline residue, and a C-terminal catalytic HECT domain[5,8] (see Fig. 1). The N-terminal C2 domain enables $Ca^{2+}$-dependent binding to membrane phospholipids, whereas the WW domains specifically bind short protein motifs, either PPXY or LPXY (where X indicates any amino acid), as well as proline-rich motifs of substrate proteins[9]. Through its four WW domains, Nedd4-2 may, nevertheless, interact with many different proteins, and even with several proteins simultaneously, because these domains show different substrate specificities, suggesting distinctive roles[10]. HECT domain is a bilobed domain whose N-terminal N-lobe interacts with E2 enzymes and whose C-terminal C-lobe contains the catalytic cysteine ($Cys^{942}$ in Nedd4-2). The C-lobe can freely move around the flexible joint loop connecting this lobe to the N-lobe in the L-shaped structure[11,12]. Furthermore, inter- or intra-molecular interactions between WW domains and the PY motif ($L^{948}PPY^{951}$) located within the HECT domain likely inhibit Nedd4-2 auto-ubiquitination, thus increasing its stability[13]. Accordingly, this interaction is disrupted by substrate binding, promoting Nedd4-2 self-ubiquitination and subsequent degradation. As a result, Nedd4-2 is downregulated upon target ubiquitination.

Nedd4-2 is also regulated by phosphorylation in response to changes in $Na^+$ or in volume through several hormonal signaling pathways. The first pathway is initiated by aldosterone, which induces SGK1 kinase-mediated Nedd4-2 phosphorylation on three sites ($Ser^{342}$, $Thr^{367}$ and $Ser^{448}$), and the second by vasopressin, which activates PKA kinase and phosphorylates Nedd4-2 on the same three residues[14–17]. Moreover, insulin signaling activates both Akt1 and SGK1 kinases and results in phosphorylation of Nedd4-2[11,18]. In particular, $Ser^{448}$ is also phosphorylated by IKKβ kinase in association with SGK or PKA kinases, inhibiting Nedd4-2 binding to ENaC upon dual phosphorylation[19]. As expected in this context, Nedd4-2 $Ser^{448}$ phosphorylation triggers 14-3-3 protein binding, which in turn inhibits the interaction between Nedd4-2 and its substrate ENaC[20,21]. In fact, the 14-3-3 protein (eta isoform) is a known cofactor in SGK- and PKA-dependent regulation of human Nedd4-2[20–22]. Concurrently, the region containing $Ser^{448}$, located between the WW2 and WW3 domains, is conserved among various Nedd4-2 proteins, thus further supporting its importance for Nedd4-2 regulation. Further evidence on the other two phosphoserines, $pSer^{342}$ and $pSer^{367}$, located in the linker between WW1 and WW2 domains, demonstrates their role as additional 14-3-3 binding motifs[17,22,23]. Several 14-3-3 binding partners also have two or more 14-3-3 binding motifs for high-affinity binding to both protomers within the 14-3-3 dimer[24,25]. However, the contribution of these individual motifs, especially of their phosphoserines $pSer^{342}$ and $pSer^{367}$, to the stability of the Nedd4-2:14-3-3 complex and the mechanism whereby 14-3-3 binding modulates Nedd4-2 function remain elusive. Bridging these knowledge gaps may enable us to develop strategies for targeted modulation of Nedd4-2 functions. Thus, to enhance our understanding of the 14-3-3-mediated Nedd4-2 regulation, we prepared and biophysically and structurally characterized the 14-3-3 binding motifs of Nedd4-2 and two longer Nedd4-2 variants (Nedd4-2$^{186–975}$ and Nedd4-2$^{335–455}$, Fig. 1) in complex with the 14-3-3 protein.

## Results

### Phosphorylated Nedd4-2 forms a stable complex with 14-3-3 with a 1:2 stoichiometry.

To investigate the interaction between Nedd4-2 and 14-3-3 proteins, we prepared a full length Nedd4-2 construct (residues 1-975), but recombinant expression and purification of this construct yielded an insoluble protein. We then expressed N-terminally truncated Nedd4-2, missing the C2 domain (Nedd4-2$^{186–975}$), which was soluble and stable enough for biophysical and structural characterization upon purification. Purified Nedd4-2$^{186–975}$ was phosphorylated by PKA in vitro, and stoichiometric phosphorylation of all three known 14-3-3 binding motifs ($Ser^{342}$, $Thr^{367}$ and $Ser^{448}$) was confirmed by LC-MS analysis[17,23]. Using the auto-ubiquitination assay[13], we also tested the catalytic activity of Nedd4-2$^{186–975}$, which was reduced in the presence of the 14-3-3η protein (Supplementary Fig. S1).

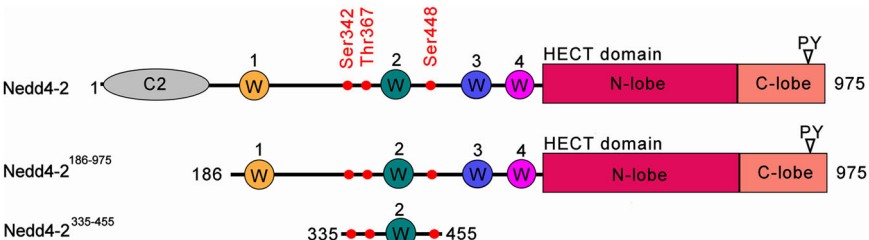

**Fig. 1 Domain structure of human Nedd4-2 and the expression constructs used in this study.** On the top, a schematic representation of the Nedd4-2 domain structure shows the relative positions of sites phosphorylated by PKA in vitro in red dots. $Ser^{342}$, $Thr^{367}$ and $Ser^{448}$ are 14-3-3 binding motifs. The $Ca^{2+}$/lipid binding domain is shown in grey (denoted as C2), and the WW domains 1-4 (denoted as W) are shown in yellow, teal, blue and magenta. The HECT domain N- and C-lobes are shown in raspberry and salmon, respectively. The boundaries of two constructs used in this study are Nedd4-2$^{186–975}$ and Nedd4-2$^{335–455}$.

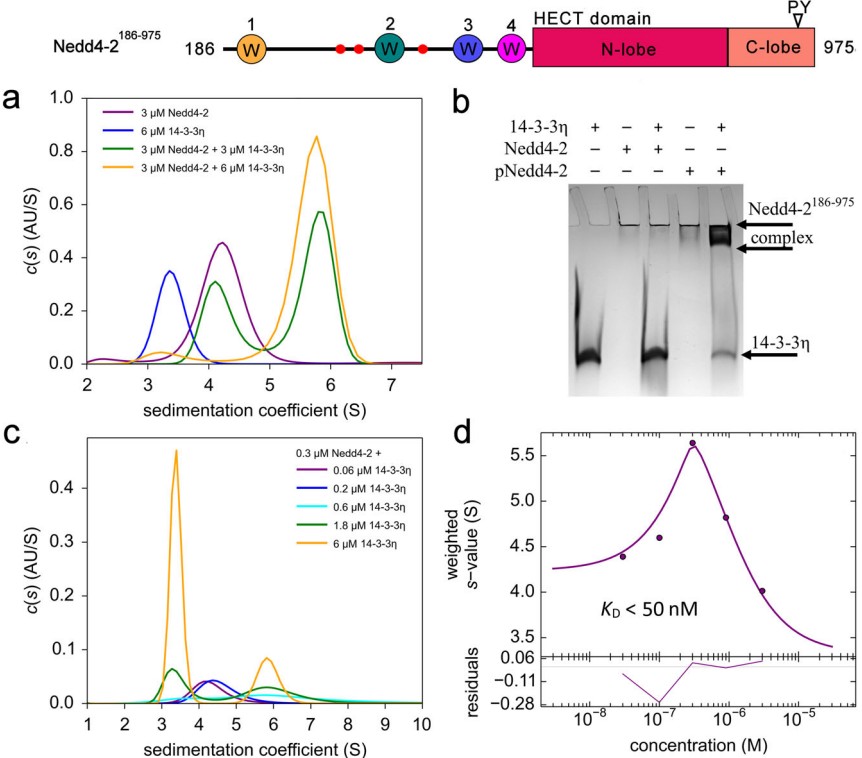

**Fig. 2 Characterization of the interaction between Nedd4-2[186−975] and 14-3-3 in solution. a** Continuous sedimentation coefficient distributions ($c(s)$) of 3 μM pNedd4-2[186-975] alone (purple), 6 μM 14-3-3η alone (blue), and pNedd4-2[186-975]:14-3-3η complex mixed at 1:1 (green) and 1:2 (yellow) molar ratios. **b** 12% TBE-PAGE showing the phosphorylation-dependent formation of a complex between pNedd4-2[186-975] and 14-3-3η after loading 240 pmol of 14-3-3η and 120 pmol of Nedd4-2[186-975] or pNedd4-2[186-975] on the native gel, respectively. **c** Sedimentation coefficient distributions ($c(s)$) of mixtures of 300 nM pNedd4-2[186-975] with 0.06−6 μM 14-3-3η. **d** Isotherm of weight-averaged sedimentation coefficients ($s_w$) derived from SV-AUC analysis of mixtures of 300 nM pNedd4-2[186-975] with 0.06−6 μM 14-3-3η. Based on our estimates, the $K_D$ value was lower than 50 nM, as further confirmed by global modeling.

Solution properties of Nedd4-2[186−975] and its interactions with 14-3-3 were characterized by sedimentation velocity analytical ultracentrifugation (SV AUC). The continuous sedimentation coefficient distributions ($c(s)$) of phosphorylated Nedd4-2[186−975] (pNedd4-2[186−975]) and 14-3-3η alone, a known Nedd4-2 binding partner[21], revealed single peaks with weight-averaged sedimentation coefficients corrected to 20.0 °C and to the density of water ($s_{w(20,w)}$) of 4.8 S (f/f₀ = 1.6) and 3.8 S (f/f₀ = 1.4), respectively (Fig. 2a). The $s_{w(20,w)}$ of pNedd4-2[186−975] and 14-3-3η correspond to $M_w$ of ~95.9 kDa and ~57.4 kDa, respectively, suggesting that pNedd4-2[186−975] is protomeric in solution (theoretical $M_w$ = 91.7 kDa), whereas 14-3-3η forms stable dimers (theoretical $M_w$ = 56.8 kDa), as expected. The analysis of $c(s)$ distributions of pNedd4-2[186−975]:14-3-3η mixtures also revealed the formation of a stable complex with a $s_{w(20, w)}$ of 6.5 S (f/f₀ = 1.6), which corresponds to a $M_w$ of ~145 kDa, thus indicating a molar stoichiometry of 1:2 (a protomer of pNedd4-2[186−975] bound to a dimer of 14-3-3η, with a theoretical $M_w$ of 149 kDa) (Fig. 2a). The formation of this complex was also confirmed by native TBE-PAGE, which showed the phosphorylation-dependent interaction between Nedd4-2[186−975] and 14-3-3η (compare lane 3 and 5) (Fig. 2b). To determine the apparent dissociation constant ($K_D$) of the pNedd4-2[186−975]:14-3-3η complex, five mixtures of pNedd4-2[186−975] and 14-3-3η were prepared at different molar ratios (from 5:1 to 1:20) and analyzed by SV-AUC (Fig. 2c). Based on our direct modeling of SV-AUC data using the Lamm equation and on our analysis of the isotherm of weight-averaged sedimentation coefficient $s$ values ($s_w$ isotherm) as a function of 14-3-3η concentration, the apparent $K_D$ is lower than 50 nM

when using a Langmuir binding model assuming a reversible interaction between the 14-3-3η dimer and one molecule of pNedd4-2[186−975] (Fig. 2d).

To examine the role of the HECT domain in the overall stability of the Nedd4-2:14-3-3η complex, we also prepared a shorter variant Nedd4-2[335−455] missing both the C2 and the HECT domains. This construct was considerably more soluble and stable than Nedd4-2[186−975]. In the SV-AUC analysis of phosphorylated Nedd4-2[335−455] (pNedd4-2[335−455]) alone, the single peak with a $s_{w(20,w)}$ of 1.4 (f/f₀ = 1.5) corresponds to a $M_w$ of ~14 kDa (theoretical $M_w$ 14 kDa). Accordingly, the absence of HECT domain should not affect Nedd4-2 oligomerization (Supplementary Fig. S2a). In turn, the analysis of pNedd4-2[335−455] and 14-3-3η mixtures showed the formation of a complex with a $s_{w(20,w)}$ of 4.5 S (f/f₀ = 1.4), which corresponds to a $M_w$ of ~70 kDa and matches a molar stoichiometry of 1:2 (a protomer of pNedd4-2[335−455] bound to a 14-3-3η dimer, theoretical $M_w$ 70 kDa) (Supplementary Fig. S2a). Both direct modeling of SV-AUC data (Supplementary Fig. S3a) and analysis of the $s_w$ isotherm as a function of pNedd4-2[335−455] concentration revealed an apparent $K_D$ lower than 30 nM when using a binding model assuming a reversible interaction between the 14-3-3η dimer and one molecule of pNedd4-2[335−455] (Fig. 3b). These results indicate that the HECT domain is not essential for high-affinity Nedd4-2 binding to 14-3-3η.

In addition, we prepared pNedd4-2[335−455] to characterize 14-3-3 isoform binding specificity. As shown by native TBE-PAGE, pNedd4-2[335−455] forms stable complexes with all human 14-3-3 protein isoforms (Supplementary Fig. S2b). Our findings

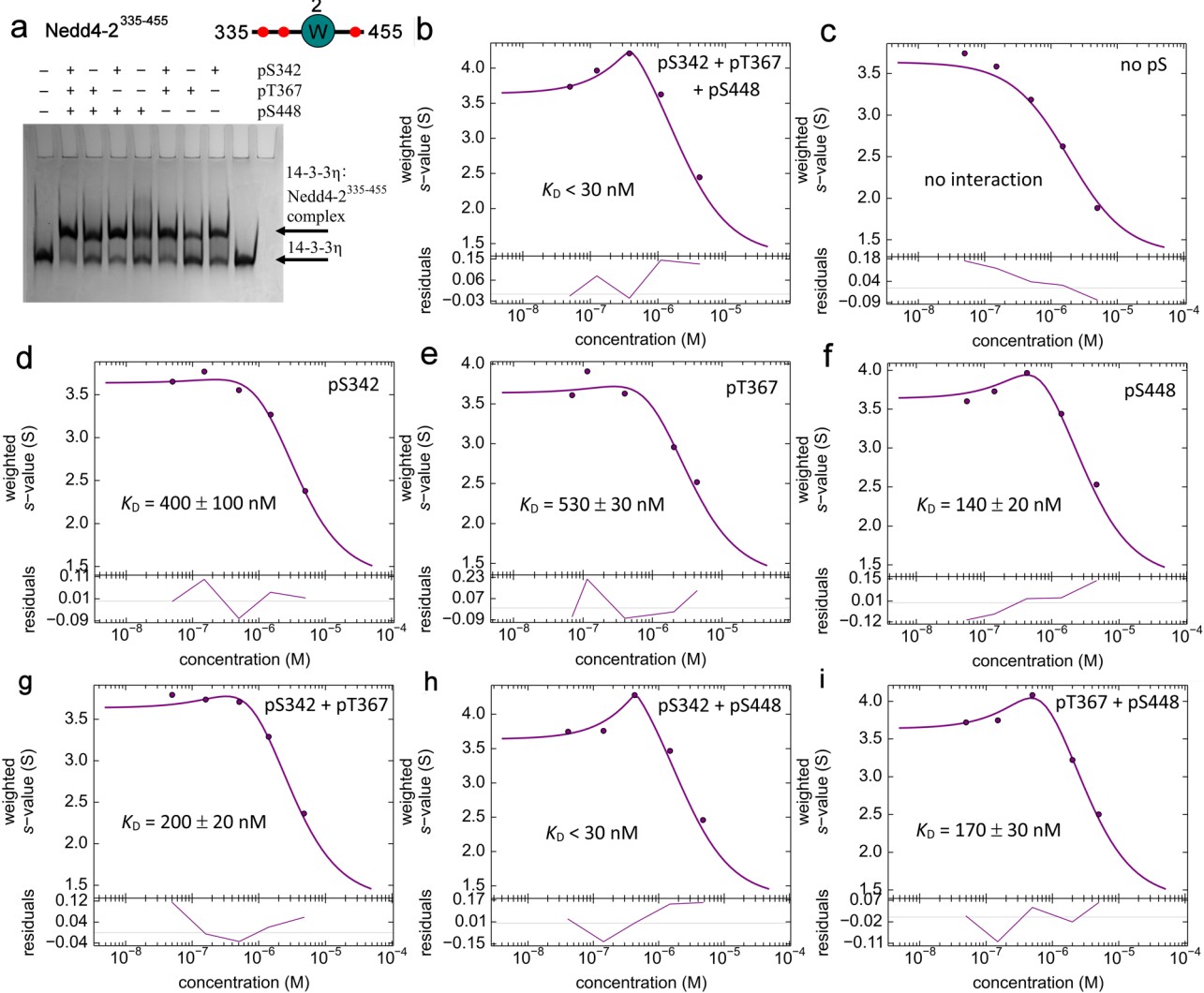

**Fig. 3 Simultaneous phosphorylation of two 14-3-3 binding motifs pSer342 and pSer448 is required for tight complex formation between pNedd4-2$^{335-455}$ and 14-3-3η. a** 12% Native TBE-PAGE showing the interaction between 14-3-3η (240 pmol) and Nedd$^{335-455}$ variants without (no pS) or with one, two or three phosphorylation sites (120 pmol); 14-3-3η protein alone was loaded on the penultimate lane. **b–i** Sedimentation velocity analytical ultracentrifugation analysis of the complexes between 14-3-3η and pNedd4-2$^{335-455}$ variants showing the $s_w$ isotherms of 14-3-3η and Nedd4-2 with all three phosphorylation sites pSer$^{342}$ + pThr$^{367}$ + pSer$^{448}$ (**b**), with no phosphorylation sites (**c**), with one phosphorylation site (pSer$^{342}$ (**d**), pThr$^{367}$ (**e**) and pSer$^{448}$ (**f**)), or with two phosphorylation sites (pSer$^{342}$ + pThr$^{367}$ (**g**), pSer$^{342}$ + pSer$^{448}$ (**h**) and pThr$^{367}$ + pSer$^{448}$ (**i**)). The isotherms of weight-averaged sedimentation coefficients were constructed by SV-AUC analysis of mixtures of 1 μM 14-3-3η with Nedd$^{335-455}$ variants (0.05 – 5 μM). The c(s) distributions underlying the $s_w$ data points are shown in Supplementary Fig. S3.

corroborate previous studies, which demonstrated that Nedd4-2 interacts with the 14-3-3 σ and η isoforms[20,21,26] and with the 14-3-3β,ε heterodimer[27].

**High-affinity Nedd4-2 binding to 14-3-3η requires both Ser$^{342}$ and Ser$^{448}$ phosphorylation**. All three putative 14-3-3 binding motifs of Nedd4-2 contain arginine residues at -5 and -3 positions with the respect to the phosphoresidue and a serine residue at -2 position. However, only the motif with Ser$^{448}$ has a proline residue at +2 position, thus resembling the canonical 14-3-3 binding motif I (RSXpSXP). Considering their differences in primary structure, these three motifs likely differ in their binding affinities to 14-3-3[25,28,29]. To characterize the importance of these phosphorylated motifs for Nedd4-2 binding to 14-3-3η, we prepared three Nedd4-2$^{335-455}$ mutants containing a single phosphorylation site (denoted as pS342, pT367 and pS448) by mutating the other phosphorylation sites to alanine and three Nedd4-2$^{335-455}$ mutants with combinations of two

phosphorylation sites (denoted as pS342 + pT367, pS342 + pS448 and pT367 + pS448), in addition to a variant with all sites mutated to alanine (denoted as no pS), which was used as a negative control.

Differences in 14-3-3η-protein binding affinity between all eight Nedd$^{335-455}$ variants (WT and mutants with none, one or two phosphosites) were tested by both native TBE-PAGE and SV-AUC analysis (Fig. 3 and Supplementary S2). Native TBE-PAGE revealed that only Nedd$^{335-455}$no pS lost its ability to bind to 14-3-3η because all other versions were able to form the complex (Fig. 3a). Among these variants, Nedd$^{335-455}$WT (Fig. 3b) and Nedd$^{335-455}$pS342 + pS448 (Fig. 3h) had the highest binding affinity (with $K_D$ lower than 30 nM), as shown in a more quantitative analysis by SV-AUC. The results from our analytical ultracentrifugation analysis also highlighted that Nedd4-2:14-3-3η-protein binding affinity was partly reduced ($K_D$ = 140 −200 nM) in singly phosphorylated Nedd$^{335-455}$pS448 (Fig. 3f), doubly phosphorylated Nedd$^{335-455}$pS342 + pT367 (Fig. 3g) and

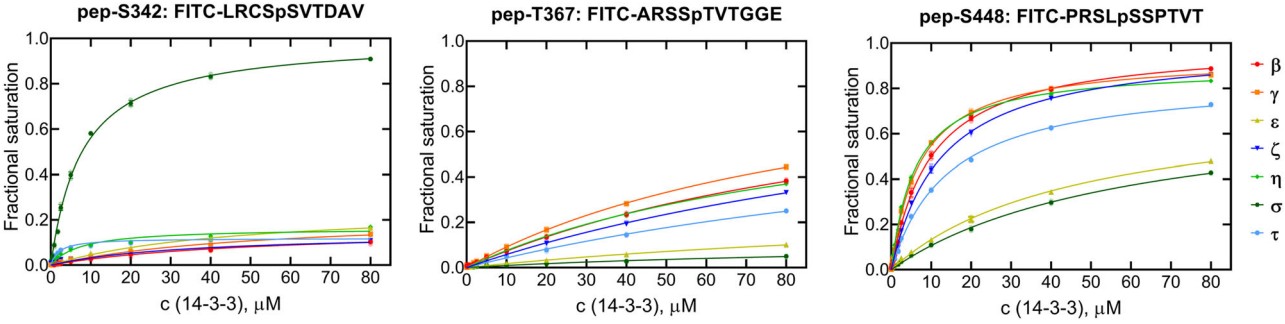

**Fig. 4 14-3-3 isoform binding specificity of Nedd4-2 phosphomotifs.** Binding of FITC-labeled phosphopeptides with 14-3-3 binding motifs of Nedd4-2 (pSer$^{342}$, pThr$^{367}$ or pSer$^{448}$) to all human 14-3-3 isoforms was characterized by FP titrations. The binding affinities of these peptides were determined by fitting FP data to a one-site binding model. The corresponding sequences of these peptides are shown at the top. Background polarization was subtracted from all values. Results are expressed as mean ± SD (n = 3). Error bars are not visible if they are smaller than the size of the symbol.

**Table 1 14-3-3 isoform binding specificity of individual 14-3-3 binding motifs of Nedd4-2.**

| | $K_D^1$, μM pep-S342 | $K_D^1$, μM pep-S448 |
|---|---|---|
| 14-3-3β | — | 10.0 ± 1.0 |
| 14-3-3γ | — | 7.2 ± 0.7 |
| 14-3-3ε | — | 50.0 ± 10.0 |
| 14-3-3ζ | — | 13.0 ± 1.0 |
| 14-3-3η | — | 6.1 ± 0.5 |
| 14-3-3σ | 7.8 ± 0.8 | 60.0 ± 10.0 |
| 14-3-3τ | — | 14.0 ± 1.0 |

$^1$The apparent $K_D$ values of the interaction between the peptide and 14-3-3 protein isoforms were determined by fluorescence polarization measurements of FITC-labeled Nedd4-2 peptides pep-S342 and pep-S448 peptides titrated with 14-3-3.

Nedd$^{335-455}$pT367 + pS448 (Fig. 3i), significantly reduced ($K_D = 400 ± 100$ nM and $530 ± 30$ nM, respectively) in the singly phosphorylated variants Nedd$^{335-455}$pS342 (Fig. 3d) and Nedd$^{335-455}$pT367 (Fig. 3e) and undetected in Nedd$^{335-455}$no pS (Fig. 3c). The binding affinities of our singly phosphorylated variants correlate with previous observations of their relative contribution to the cAMP-dependent regulation of Nedd (Ser$^{448}$ > Ser$^{342}$ > Thr$^{367}$)[17]. Therefore, bidentate interaction through pSer$^{342}$ and pSer$^{448}$ affords the most stable complex between Nedd4-2 and 14-3-3η in which pSer$^{448}$ is the high affinity motif[30].

**14-3-3 binding motifs of Nedd4-2 show distinct 14-3-3 isoform binding specificity.** The interactions between isolated 14-3-3 binding motifs of Nedd4-2 and 14-3-3 protein isoforms were further characterized by fluorescence polarization (FP). For this purpose, we synthesized three decapeptides of known 14-3-3 binding motifs with phosphorylated pSer$^{342}$, pThr$^{367}$ and pSer$^{448}$ (denoted as pep-S342, pep-T367 and pep-S448) bordered by the amino acids from the natural sequence of Nedd4-2, with four amino acids before and another five after the corresponding phosphoresidue. We labelled all peptides N-terminally with fluorescein isothiocyanate (FITC) and characterized their binding to all human 14-3-3 protein isoforms by FP (Fig. 4 and Table 1). Our FP data revealed that all three phosphopeptides have relatively low binding affinities. The pep-S342 showed a measurable binding affinity only to 14-3-3σ ($K_D$ of ~7.8 μM). In contrast, we were able to detect pep-T367 binding, albeit very weak, to all 14-3-3 isoforms except ε and σ. Lastly, we determined that pep-S448 binds to all 14-3-3 isoforms in the low micromolar range, showing the lowest $K_D$ values (~6.1 and 7.2 μM) when bound to isoforms η and γ, respectively. In addition, the $K_D$ values of

synthetic phosphorylated peptides are two orders of magnitude higher than those of recombinant Nedd4-2 variants, indicating simultaneous binding of two motifs and/or the participation of regions other than phosphorylated motifs in the formation of the complex (Figs. 2 and 3). Overall, these results support our hypothesis that the pSer$^{448}$ motif is the dominant 14-3-3 binding motif.

**The 14-3-3 dimer simultaneously anchors both pSer$^{342}$ and pSer$^{448}$ motifs of Nedd4-2.** Because our AUC analysis suggested that the pSer$^{342}$ and pSer$^{448}$ motifs are responsible for high-affinity Nedd4-2 binding to 14-3-3, we investigated the structural basis of their interaction with 14-3-3 in detail. For this purpose, we crystallized the peptides containing the pSer$^{342}$ and pSer$^{448}$ motifs (pep-S342 and pep-S448) bound to 14-3-3γΔC (lacking the 15 flexible residues at the C-terminus). The 14-3-3γ isoform was selected based on crystal quality. Both phosphopeptide complexes crystallized in the trigonal space group R3 with two 14-3-3 dimers with bound phosphopeptides in the asymmetric unit (PDB ID: 6ZBT and 6ZC9, Table 2 and Fig. 5). The structures of the pep-S342:14-3-3γΔC and pep-S448:14-3-3γΔC complexes were solved by molecular replacement with 14-3-3γ (PDB ID: 2B05) as a search model and refined to resolutions of 1.8 Å and 1.9 Å, respectively. The final electron densities of both peptides allowed us to trace seven of ten amino acids (Leu$^{338}$ – Thr$^{344}$) of pep-S342 and (Pro$^{444}$ – Pro$^{450}$) pep-S448 (Fig. 5a, c). In both cases, we were unable to trace the last three residues of each peptide, presumably due to disorder. The phosphate group, the main-chain conformation and other contacts in the 14-3-3-binding groove of both motifs were recognized similarly to those previously observed in other 14-3-3 protein complexes[25,29,31–34]. The pSer$^{342}$ moiety of pep-S342 is coordinated through direct contacts enabled by side chains of the 14-3-3 residues Arg$^{57}$, Arg$^{132}$, Lys$^{50}$ and Tyr$^{133}$ and by water-mediated contacts with 14-3-3 Asp$^{129}$ and Asn$^{178}$ (Fig. 5b). Other contacts include hydrogen bonds between the main-chain atoms of the Nedd4-2 residues Cys$^{341}$ and Val$^{343}$ and the side chains of the 14-3-3 residues Asn$^{229}$ and Asn$^{178}$, respectively. In addition, the side chain of Nedd4-2 Ser$^{340}$ makes a polar contact with the side-chain residues Trp$^{233}$ and Glu$^{185}$ of 14-3-3. Similar contacts between the phosphopeptide and 14-3-3 were also observed in the 14-3-3γΔC:pep-S448 complex (Fig. 5d). The main difference is the absence of water-mediated contacts between Arg$^{445}$ (-5 residue with the respect to pSer$^{448}$) and the side chains of the 14-3-3 residues Arg$^{57}$, Arg$^{61}$ and Glu$^{136}$.

Because the 14-3-3 binding motifs pSer$^{342}$ and pSer$^{448}$ border the WW2 domain (Fig. 1), we assessed whether this domain

**Table 2 Crystallographic data collection and refinement statistics.**

| Complex | pep-S342:14-3-3γΔC | pep-S448:14-3-3γΔC | Nedd4-2[335–455]:14-3-3ηΔC |
|---|---|---|---|
| PDB ID | 6ZBT | 6ZC9 | 7NMZ |
| Wavelength (Å) | 0.9184 | 0.9184 | 1.3418 |
| Space group | R 3:H | R 3:H | C121 |
| Unit-cell parameters | | | |
| $a$, $b$, $c$ (Å) | 205.857 205.857 74.354 | 205.707 205.707 74.649 | 117.86 58.95 106.76 |
| α, β, γ (°) | 90.0 90.0 90.0 | 90.0 90.0 120.0 | 90 90.693 90 |
| Asymmetric unit contents | Dimer of 14-3-3γΔC with bound phosphopeptide | Dimer of 14-3-3γΔC with bound phosphopeptide | Dimer of 14-3-3ηΔC with bound Nedd4-2 containing pSer[342] and pSer[448] |
| Resolution range (Å)[a] | 27.27 – 1.799 (1.864 – 1.799) | 25.53 – 1.899 (1.967 – 1.899) | 31.35 – 2.303 (2.385 – 2.303) |
| Unique reflections | 108872 (10874) | 92751 (9202) | 32696 (3215) |
| Data multiplicity | 5.81 (8.77) | 5.84 (9.45) | 4.73 (3.99) |
| Completeness (%) | 99.88 (99.65) | 99.76 (98.90) | 99.85 (99.38) |
| $\langle I/\sigma(I)\rangle$ | 25.82 (2.9) | 20.39 (1.81) | 19.68 (1.94) |
| $R_{meas}$[b] | 0.041 (0.597) | 0.051 (1.038) | 0.064 (0.655) |
| $R_{work}$ | 0.2101 (0.2973) | 0.2119 (0.3793) | 0.1994 (0.2432) |
| $R_{free}$[c] | 0.2404 (0.3240) | 0.2394 (0.3915) | 0.2350 (0.2915) |
| No. of protein atoms | 7332 | 7393 | 3718 |
| No. of ligand atoms | 40 | 40 | |
| No. of waters | 642 | 551 | 189 |
| Average B factors (Å²) | 36.20 | 44.80 | 43.92 |
| Protein | 35.65 | 44.47 | 44.03 |
| Ligand | 49.35 | 57.13 | |
| Water | 41.72 | 48.36 | 41.82 |
| R.m.s.[d] deviations from ideal values | | | |
| Bond lengths (Å) | 0.003 | 0.004 | 0.003 |
| Bond angles (°) | 0.50 | 0.61 | 0.61 |
| Ramachandran favored (%) | 99.12 | 98.90 | 98.73 |
| Ramachandran allowed (%) | 0.88 | 1.10 | 1.27 |
| Ramachandran outliers (%) | 0 | 0 | 0 |

[a]Values in parentheses are for the highest resolution shell.
[b]$R_{meas} = \sum_{hkl} \{N(hkl)/[N(hkl) - 1]\}^{1/2} \times \sum_i |I_i(hkl) - \langle I(hkl)\rangle| / \sum_{hkl} \sum_i I_i(hkl)$, where $I(hkl)$ is the intensity of reflection hkl, $\langle I(hkl)\rangle = \frac{1}{N(hkl)}\sum_i I_i(hkl)$, and $N(hkl)$ the multiplicity.
[c]The free R value ($R_{free}$) was calculated using 5% of the reflections, which were omitted from the refinement.
[d]R.m.s., root mean square.

participates in 14-3-3 binding by crystallizing the Nedd4-2[335–455] construct in a complex with 14-3-3ηΔC. The phosphorylation site Thr[367] was mutated to Ala to prepare Nedd4-2[335–455] phosphorylated only at Ser[342] and Ser[448] for high-affinity binding and for homogeneous crystallization. The 14-3-3 ηΔC:Nedd4-2[335–455] complex crystallized in the monoclinic space group C121, with one Nedd4-2 molecule bound to the 14-3-3ηΔC dimer in the asymmetric unit. The structure was solved by molecular replacement with 14-3-3η (PDB ID: 2C63) as a search model and refined to a resolution of 2.3 Å. Interpretable electron density was found for 21 residues, nine around the pSer[342] site and twelve around second pSer[448] site (Leu[338]-Ala[346] and Pro[444]-Ser[455]) (PDB ID code 7NMZ, Fig. 5e, Table 2). Although the structure revealed simultaneous binding to both phosphorylated motifs by a 14-3-3η dimer, no electron density was found for the 97 residues between the phosphorylated motifs (Val[347]-Arg[443]), thus indicating that this region, including the WW2 domain, remains flexible when bound to 14-3-3 (Fig. 5e). The recognition of the phosphate group and contacts with the 14-3-3 ligand binding grooves are similar to those observed in the structures of the individual motifs (Fig. 5b, d). Other contacts were not detected in the crystal structures of short phosphopeptides, including the 14-3-3 residue Asn[43] from the α3 helix of both protomers, which interacts with the main chain of the Nedd4-2 residues Ala[346], Val[452], Thr[453] and Ser[455], and the polar contacts between the 14-3-3 residue Glu[15] from the α1 helix and the Nedd4-2 residues Thr[453] and Ser[455]. A similar contacts have been previously observed in the structures of the Nth1:14-3-3 and caspase-2:14-3-3 complexes (PDB ID: 5N6N[32] and 6SAD[33], respectively).

**14-3-3η interacts with the WW3 and HECT domains of Nedd4-2 and changes the relative positions of all Nedd4-2 domains.** Nedd4-2[186–975] chemical crosslinking using the bifunctional agent disuccinimidyl glutarate (DSG) coupled to MS (XL-MS) with and without 14-3-3η showed further 14-3-3 protein-mediated structural changes in Nedd4-2, highlighting 14 and 13 intramolecular distance constraints for Nedd4-2[186–975] in the free and complexed forms, respectively (Supplementary Table S1, S2, Fig. 6a, b). Many crosslinks were observed in both forms of Nedd4-2[186–975], for example, the crosslink between Lys[598] from the HECT domain and Thr[275] from the WW1-WW2 linker (#4 in Supplementary Table S1, #6 in Supplementary Table S2). In other words, when the complex is formed, the distance between these two regions remains unaltered (Fig. 6a, b). In contrast, crosslinks connecting the WW2 domain to the WW1-WW2 linker (Lys[395]-Thr[275], #6 in Supplementary Table S1) and WW2 to the HECT domain (Lys[398]-Lys[822] and Lys[398]-Ser[932], #8 and 14 in Supplementary Table S1) were observed only in the free form of Nedd4-2[186–975]. Conversely, crosslinks connecting the WW2-WW3 and WW3-WW4 linkers (Ser[538]-Ser[428], #3 in Supplementary Table S2) and WW3 with the C-lobe of HECT (Lys[531]-Lys[935], #13 in Supplementary Table S2) were observed only in the complexed form of Nedd4-2[186–975]. These findings confirm that the formation of this complex affects the position of the WW2 domain with respect to the HECT domain and other WW domains of Nedd4-2[186–975].

To quantify intramolecular Nedd4-2[186–975] crosslinks with and without 14-3-3η, we crosslinked both Nedd4-2[186–975] alone and the Nedd4-2[186–975]:14-3-3η complex using [12]C and [13]C-labeled

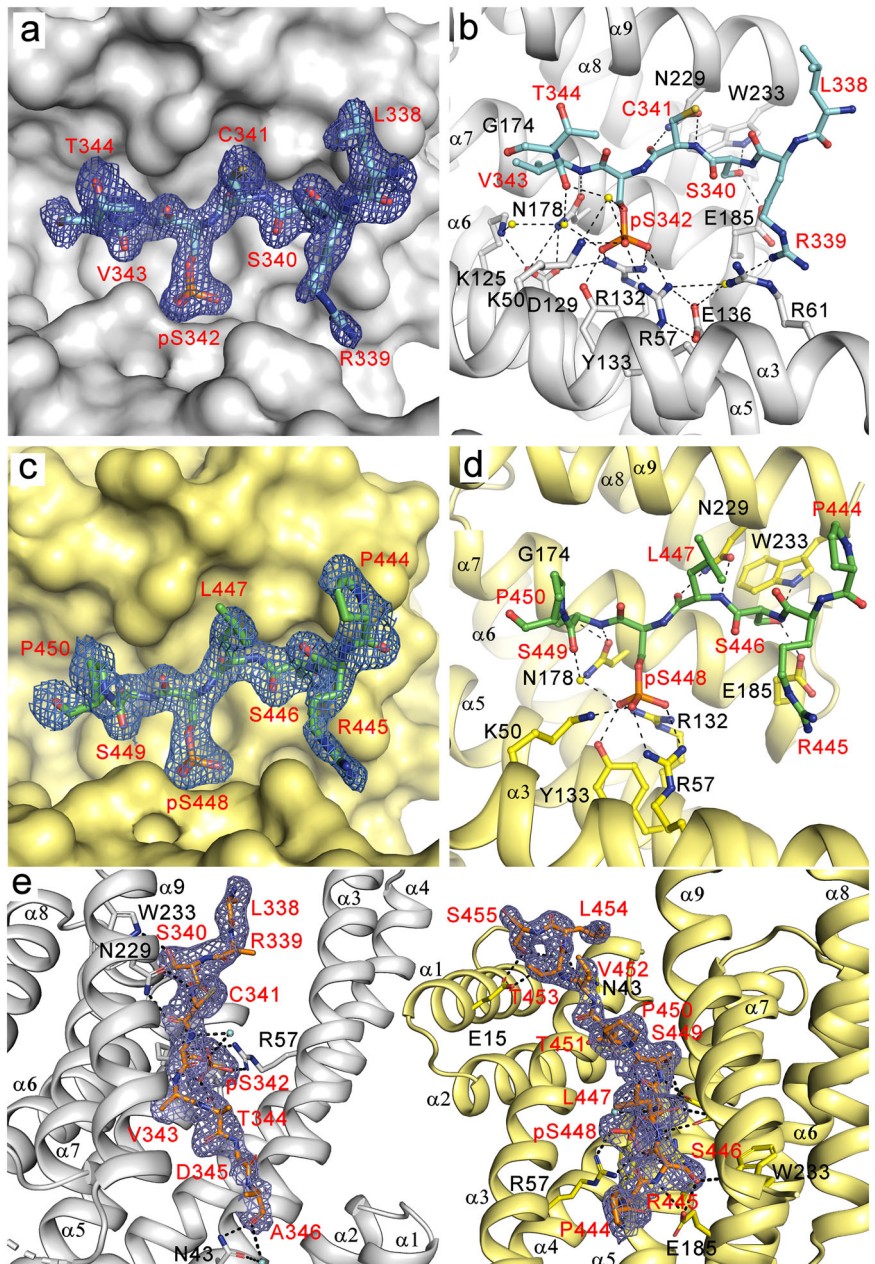

**Fig. 5 Crystal structures of Nedd4-2 peptides containing the 14-3-3 binding motifs pSer342 and pSer448 bound to 14-3-3γ and 14-3-3η. a** Detailed view of the crystal structure of the 14-3-3γΔC:pep-S342 complex. The *2F$_o$-F$_c$* electron density map is contoured at 1σ. **b** Polar contacts (black lines) between the residues of 14-3-3γ (black) and the pSer[342] binding site of Nedd4-2 (red). Water molecules are shown as yellow spheres. **c** Detailed view of the crystal structure of the 14-3-3γΔC:pep-S448 complex. The *2F$_o$-F$_c$* electron density map is contoured at 1σ. **d** Polar contacts (black lines) between the residues of 14-3-3γ (black) and the pSer[448] binding site of Nedd4-2 (red). Water molecules are shown as yellow spheres. **e** Crystal structure of Nedd4-2[335–455] containing two phosphorylation sites (pSer[342] and pSer[448]) bound to the 14-3-3ηΔC dimer: Top view of the intervening linker sequence of the two 14-3-3-binding motifs of Nedd4-2 and of the polar contacts (black dashed lines). The final *2F$_o$ – F$_c$* electron density map is contoured at 0.8σ (blue mesh). Nedd4-2 residues are labeled in red, and 14-3-3 residues are labeled in black. Water molecules are shown as cyan spheres. Structure figures were generated using PYMOL.

disuccinimidyl adipate (DSA) in a 1:1 ratio (Supplementary Table S3, Fig. 6a, b – in green). Upon complex formation, only the abundances of crosslinks #1 and #3 significantly changed, indicating that Nedd4-2 residues Lys[531] from the WW3 domain and Lys[607] from the HECT domain (#1 in Supplementary Table S3, Fig. 6b) are crosslinked mainly in the absence of 14-3-3η. Conversely, the crosslink between the Nedd4-2 residues His[186], which precedes the WW1 domain, and Lys[639] from the N-

terminus of the HECT domain (#3 in Supplementary Table S3, Fig. 6b) primarily formed in the presence of 14-3-3η.

Crosslinking Nedd4-2[186–975]:14-3-3η mixtures (mixed in 1:2 stoichiometry) using disuccinimidyl suberate (DSS) and DSG yielded eight intermolecular crosslinks (Supplementary Table S4, Fig, 6c). Most of these crosslinks connect the α-helices that form the 14-3-3η ligand binding groove (α3, α5 and α9) with the WW3 domain and the HECT domain

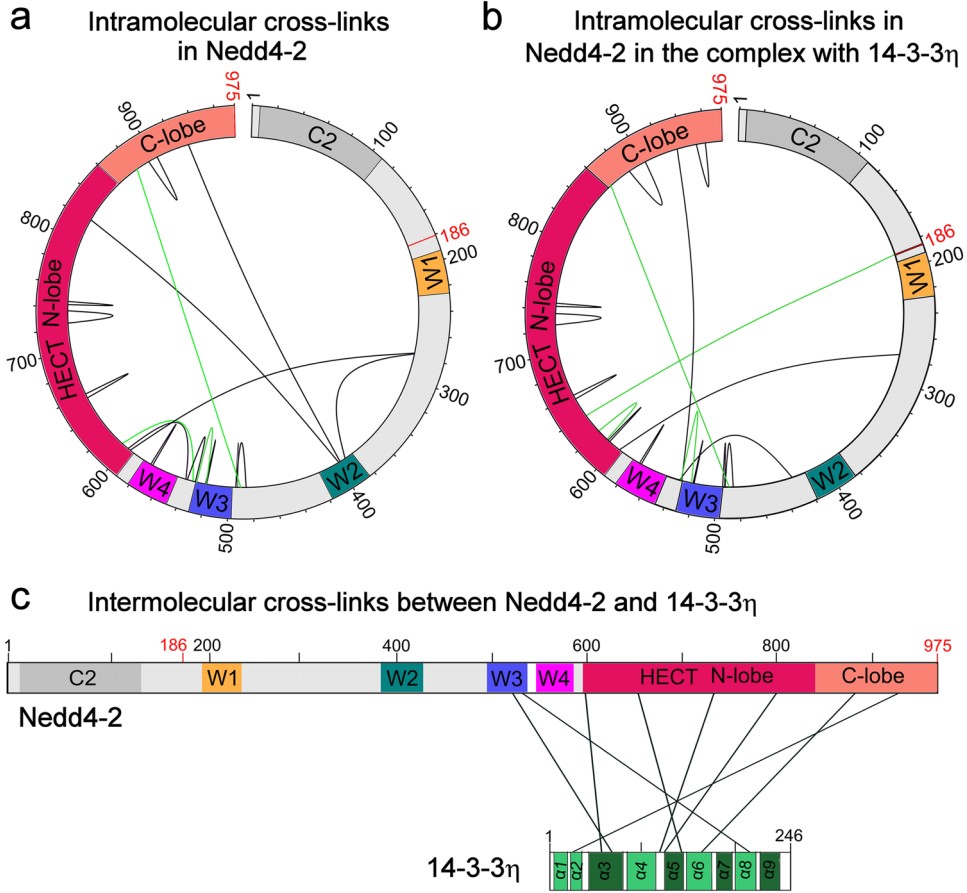

**Fig. 6 Schematic representation of intra- and intermolecular crosslinks in Nedd4-2 and between Nedd4-2 and 14-3-3η. a** Schematic representation of intramolecular crosslinks of Nedd4-2[186–975] alone (black) with a 50-fold molar excess of DSG (Supplementary Table S1) and quantitative crosslinks with a 50-fold molar excess of 12 C and 13 C DSA in pale green (Supplementary Table S3). **b** Schematic representation of intramolecular crosslinks of Nedd4-2[186–975] in a complex with 14-3-3η (black) with a 50-fold molar excess of DSG (Supplementary Table S2) and quantitative crosslinks with a 50-fold molar excess of 12 C and 13 C DSA in pale green (Supplementary Table S3). **c** Schematic representation of intermolecular crosslinks between Nedd4-2[186–975] and 14-3-3η mixed at a 1:2 stoichiometry with a 50-fold molar excess of DSG or DSS (black) crosslinking agent (Supplementary Table S4). Nedd4-2[186–975] diagram: the C2 domain, not present in the Nedd4-2[186–975] construct, is shown in grey, the WW domains (denoted as W1-4) are shown in yellow, teal, blue and magenta, and the HECT domain N- and C-lobes are shown in raspberry and salmon, respectively. The beginning and the end of the Nedd4-2[186–975] construct are labeled in red. 14-3-3 diagram: colored regions represent α-helices of the 14-3-3 molecule (shown in green), and the helices that form the ligand binding groove (α3, α5, α7 and α9) are colored in dark green. This figure was prepared using the xiVIEW (https://xiview.org/) and InkScape (http://www.inkscape.org/) programs.

(crosslinks #1-3 and #5, 6), and the 14-3-3η helices α2 and α6 with the HECT domain (crosslinks #4, 7 and 8). Most likely, these 14-3-3η and Nedd4-2 regions directly interact with each other in the complex. The combined results from our crosslinking experiments demonstrate that 14-3-3η interacts with the WW3 and HECT domains of Nedd4-2 and changes the relative positions of all Nedd4-2 domains.

**14-3-3η binding blocks interactions between WW domains and the HECT domain of Nedd4-2.** To gain structural insights into the 14-3-3-mediated regulation of Nedd4-2, we tried to crystallize the Nedd4-2[186–975]:14-3-3η complex, but all our crystallization trials were unsuccessful. For this reason, we performed size-exclusion chromatography (SEC) coupled to small angle x-ray scattering (SAXS) analysis of Nedd4-2[186-97], 14-3-3η and the Nedd4-2[186–975]:14-3-3η complex.

Scattering data from two regions of the elution peak of Nedd4-2[186–975] alone were analyzed (Supplementary Fig. S4a). Based on our Guinier analysis, the data from the top of the peak of Nedd4-2[186–975] revealed the presence of aggregates, whereas the data from the right shoulder of the elution peak showed homogenous

particles with consistent $R_g$ values of ~4.4 nm and an estimated $M_w$ of ~101 kDa, in line with the theoretical $M_w$ of Nedd4-2[186–975] (92 kDa; Supplementary Fig. S4b and Table 3). In turn, scattering data from the top of the elution peak of 14-3-3η indicated homogenous particles with an estimated $M_w$ of 56 kDa, matching the theoretical $M_w$ of 14-3-3η dimer (57 kDa) (Supplementary Fig. S4c, d and Table 3). The Nedd4-2[186–975]:14-3-3η complex was prepared with the 1:2 stoichiometry at a concentration 60 μM, which is approximately three orders of the magnitude higher than the $K_D$ value derived from SV-AUC (~50 nM, Fig. 2). The complex eluted as a single peak and frames from two regions of the elution profile were analyzed (Supplementary Fig. S4e). As in Nedd4-2[186–975], the Guinier plot of the scattering data from the top of the elution peak indicated the presence of aggregates, but the scattering data from the right side of the peak showed a $R_g$ of ~4.8 nm and a $M_w$ of ~157 kDa, corroborating the theoretical $M_w$ of the Nedd4-2[186–975]:14-3-3η complex with a 1:2 stoichiometry (149 kDa) (Supplementary Fig. S4f).

After calculating distance distribution functions $P(r)$, we found similar values of maximal distance within the particle ($D_{max}$) of

**Table 3 Structural parameters determined from SEC-SAXS data.**

| Sample | $R_g$ (nm)[c] | $R_g$ (nm)[d] | $D_{max}$ (nm) | $V_P$[e] (nm³) | $M_w$[f] (kDa) | $M_w$[g] (kDa) |
|---|---|---|---|---|---|---|
| Nedd4-2$^{186-975}$(1)[a] | 4.67 ± 0.01 | 4.92 ± 0.01 | 18.3 | 218 | 125 | 92 |
| Nedd4-2$^{186-975}$(2)[b] | 4.39 ± 0.02 | 4.54 ± 0.02 | 16.9 | 192 | 101 | 92 |
| 14-3-3η | 2.93 ± 0.01 | 2.94 ± 0.01 | 8.6 | 87 | 56 | 57 |
| 14-3-3η:Nedd4-2$^{186-975}$(1)[a] | 4.95 ± 0.01 | 5.14 ± 0.01 | 18.1 | 277 | 170 | 149 |
| 14-3-3η:Nedd4-2$^{186-975}$(2)[b] | 4.79 ± 0.01 | 4.92 ± 0.01 | 17.2 | 253 | 157 | 149 |

[a]Based on frames from the top of the elution peak from SEC.
[b]Based on frames from the right side of the elution peak from SEC.
[c]Calculated using the Guinier approximation[82].
[d]Calculated using GNOM[73].
[e]Excluded volume of the hydrated particle (Porod volume).
[f]Molecular weight estimate based on a consensus Bayesian assessment method[76].
[g]Theoretical molecular weights of proteins alone and the Nedd4-2$^{186-975}$:14-3-3η complex (with 1:2 stoichiometry).

Nedd4-2$^{186-975}$ and its complex with 14-3-3η (Fig. 7a and Table 3). Concurrently, the dimensionless Kratky plot of scattering data ($(sR_g)^2I(s)/I_0$ versus $sR_g$) suggested their conformational flexibility, as indicated by the bell-shaped profiles of the complex and Nedd4-2$^{186-975}$ alone with maxima of 1.2 at $sR_g \sim 2.1$ and 1.4 at $sR_g \sim 2.2$, respectively (Fig. 7b, green and red trace) because the scattering data on a compact globular particle such as 14-3-3η (Fig. 7b, blue trace) peaks at 1.104 at $sR_g$ value of ~1.73. Nevertheless, the complex exhibits a lower conformational flexibility than Nedd4-2$^{186-975}$ alone.

By combining rigid body modeling of SAXS profiles with distance constraints assessed by crosslinking coupled to MS (Supplementary Tables S1-S4), we calculated models of Nedd4-2$^{186-975}$ alone and in a complex with 14-3-3η. The best-scoring CORAL model of Nedd4-2$^{186-975}$ alone fitted the experimental SAXS data with a $\chi^2$ of 1.26 (Supplementary Fig. S5) and matched all intramolecular crosslinks (Supplementary Table S1 and S3). In this model, the WW2 and WW3 domains (in teal and blue, respectively) are positioned close to the HECT domain, WW2 is located between the N- and C-lobes and WW3 interacts with the C-lobe of the HECT domain (Fig. 7c). In turn, WW1 and WW4 are distant from the catalytic domain (Supplementary Fig. S5). This arrangement corroborates a previous report by Grimsey at el.[35]. According to these authors, in the closed, autoinhibited form of Nedd4-2, the HECT domain interacts with the region preceding the WW3 domain.

The Nedd4-2$^{186-975}$:14-3-3η complex was first modeled as a 14-3-3η dimer attached to Nedd4-2 via pSer$^{342}$ and pSer$^{448}$-containing motifs and assuming that the whole HECT domain is a rigid body. However, the models derived from these simulations did not fit well the experimental SAXS data and had $\chi^2$ values of ~6.4. Therefore, we subsequently allowed the N- and C-lobes of HECT to move freely with respect to each other. From these simulations, the best-scoring CORAL model (Supplementary Fig. S6) showed a considerably better agreement with the experimental SAXS data ($\chi^2$ of 1.63), matching all intermolecular crosslinks (Supplementary Table S4), the intramolecular cross-links of complexed Nedd4-2 (Supplementary Table S2 and S3) and the ab initio shape reconstruction (Supplementary Fig. S6b, S4f). The model of the complex suggests that the WW3 domain of Nedd4-2 (shown in blue) is positioned within the central channel of the 14-3-3η dimer and interacts with the 14-3-3η helices α1, α3 and α9 (Fig. 7d). The HECT domain is located outside the central channel of 14-3-3η dimer, its C-lobe (shown in salmon) is close to the 14-3-3η α8-α9 loop, and the N-lobe (shown in raspberry) is positioned near the 14-3-3η helices α6 and α8. Although the HECT domain interacts with one 14-3-3η protomer, the WW4 and WW2 domains are located above the second 14-3-3η protomer, far from the HECT domain (Supplementary Fig. S6a). Thus, by comparison with the model of Nedd4-2$^{186-975}$ alone,

14-3-3η binding changed the relative positions of the structured domains of Nedd4-2, including the N- and C-lobes of HECT, where the C-lobe interacts with the helices α6, α7 and α10 of the N-lobe. Taken together, our SAXS and crosslinking data highlight that 14-3-3 binding induces a conformational rearrangement of Nedd4-2 by changing the relative positions and interactions of its structured domains, including the N- and C-lobes of the catalytic HECT domain, and by masking the surface of the WW3 domain.

## Discussion

Nedd4-2 primarily ubiquitinates membrane-bound proteins, such as channels and receptors, and its dysfunction leads to multiple diseases, including epilepsy, hypertension, cystic fibrosis, pulmonary edema or Liddle syndrome[36–40]. Previous studies have shown that Nedd4-2 phosphorylation triggers binding to the adaptor protein 14-3-3 and that this interaction may have different functional consequences depending on the tissue. In the brain, for example, this interaction promotes the ubiquitination of the GluA1 subunit of the AMPA receptor[36]. In contrast, in the kidney, Nedd4-2 binding to 14-3-3 weakens ENaC ubiquitination, thereby increasing sodium absorption by elevating the surface expression of ENaC[22]. However, key structural aspects of the 14-3-3-mediated regulation of Nedd4-2 have remained unclear until now. Thus, our study provides detailed structural insights into the interaction between 14-3-3 protein and Nedd4-2 by combining several structural biology approaches.

The interaction between the 14-3-3 protein and Nedd4-2 is mediated by phosphorylation of three residues bordering WW2 domain: pSer$^{342}$, pThr$^{367}$ and pSer$^{448}$ (Fig. 1)[17,23]. Because 14-3-3 protein dimers frequently interact with their binding partners by simultaneously anchoring two phosphorylated motifs separated by at least 20 amino acid residues[41], we hypothesized that the interaction between 14-3-3 and Nedd4-2 is also mediated by such bidentate binding. Our LC-MS analysis confirmed that Nedd4-2 is phosphorylated by PKA in vitro on all three aforementioned sites. Mutating all three residues abolished Nedd4-2 binding to the 14-3-3 protein, whereas simultaneous phosphorylation of the Ser$^{342}$ and Ser$^{448}$ sites led to the highest affinity binding of Nedd4-2 to the 14-3-3η dimer, in the nanomolar range ($K_D < 30$ nM) (Fig. 3, S3). The binding affinity decreased upon single phosphorylation of Ser$^{448}$ or other double phosphorylation combinations involving Ser$^{448}$. Our fluorescence polarization measurements also highlighted the importance of the Ser$^{448}$-containing motif for high-affinity Nedd4-2 binding to 14-3-3η. Concurrently, our crystallographic analysis of the phosphorylated Nedd4-2$^{335–455}$ bound to 14-3-3η confirmed the bidentate interaction with the 14-3-3 dimer through both 14-3-3 binding motifs, pSer$^{342}$ and pSer$^{448}$, also showing that the complex establishes more contacts than the structures of the isolated singly phosphorylated motifs (Fig. 5). These findings support the

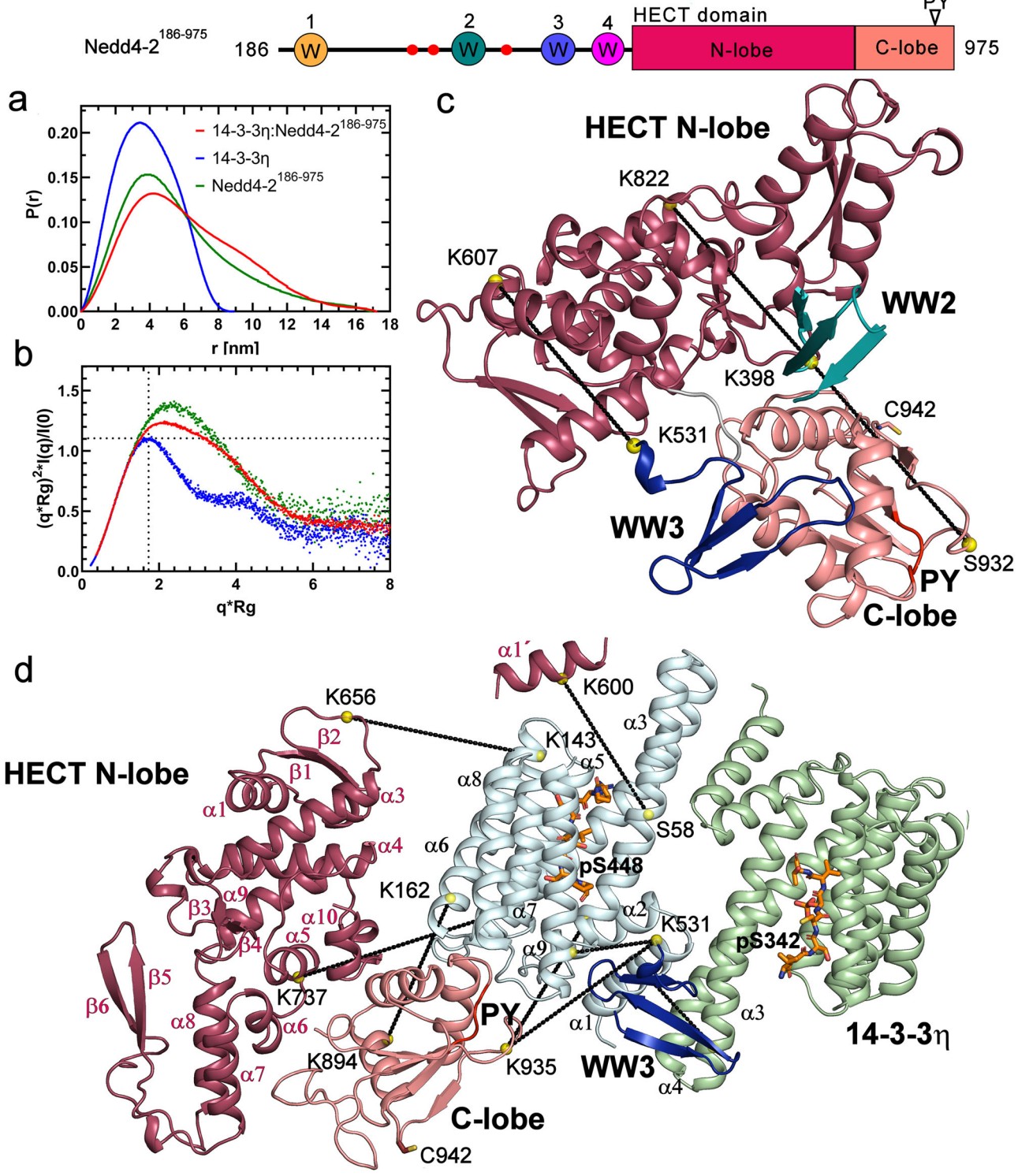

**Fig. 7 SAXS-based structural modeling of the pNedd4-2$^{186-975}$:14-3-3η complex and pNedd4-2$^{186-975}$. a** Plot of the distance distribution functions P(r), with maximum particle dimensions ($D_{max}$) of 169, 89 and 172 Å for Nedd4-2$^{186-975}$, 14-3-3η and the 14-3-3η:pNedd4-2$^{186-975}$ complex, respectively. **b** Dimensionless Kratky plots are shown in green for Nedd4-2$^{186-975}$, in blue for 14-3-3η, and in red for the 14-3-3η:pNedd4-2$^{186-975}$ complex. The dotted lines mark the value of 1.104 for $sR_g = 1.73$, denoting the peak of a perfectly globular particle. **c** Crosslinked domains from the best-scoring CORAL model of Nedd4-2$^{186-975}$. The full model is shown in Supplementary Fig. S5. **d** Crosslinked domains from the best-scoring CORAL model of the Nedd4-2$^{186-975}$:14-3-3η complex. The full model is shown in Supplementary Fig. S6. The 14-3-3η protomers are shown in pale green and pale cyan. In the HECT domain of Nedd4-2$^{186-975}$, the N-lobe is indicated in raspberry and the C-lobe in salmon. WW1, WW2, WW3 and WW4 domains are indicated in yellow, teal, blue and magenta, respectively. Phosphorylated 14-3-3 binding motifs of Nedd4-2 are shown as orange sticks (PDB ID: 6ZBT and 6ZC9, this work). Cα atoms of crosslinked residues are shown as yellow spheres. The PY motif (L$^{948}$PPY$^{951}$) is shown in red, and the catalytic residue Cys$^{942}$ is shown as a stick. The elements in the HECT N-lobe secondary structure are numbered according to[81].

assumptions that the key residue for 14-3-3 protein binding is Ser[448], phosphorylated by either SGK or PKA, and that the interaction between Nedd4-2 and 14-3-3η depends on a Pro residue located at position +2 from pSer[448] [17,20,22]. Although pSer[448] alone promotes Nedd4-2 binding to 14-3-3, our data clearly show that the most stable interaction involves simultaneous binding of two phosphorylated motifs. In line with our results, decreased phosphorylation of Nedd4-2 at Ser[342] has been recently shown to promote its association with ribosomal proteins during endoplasmic reticulum stress[42], but whether this process also involves changes in the interaction between Nedd4-2 and 14-3-3 is still unknown. Nevertheless, our structural analysis of this interaction did show that 14-3-3 changes the relative positions of all Nedd4-2 domains.

14-3-3 proteins are well known to regulate their binding partners by modulating their structure and/or masking structural and functional features on their surface. The conformational change upon the 14-3-3 protein binding was demonstrated for the serotonin N-acetyltransferase (AANAT), an enzyme that controls the daily rhythm in melatonin synthesis. The 14-3-3 binding modulates structure of the substrate binding sites of AANAT, thereby increasing the affinity of AANAT for its substrates with an accompanying increase in activity[43,44]. In our previous work on 14-3-3-mediated regulation of the neutral trehalase Nth1, we demonstrated that 14-3-3 protein triggers its enzyme activity by stabilizing the interaction between its catalytic and regulatory domains[32]. Another example of 14-3-3-induced structural changes upon binding is the negative regulation of B-RAF kinase, which 14-3-3 protein maintains in an inactive state by blocking the membrane recruitment of B-RAF and by preventing B-RAF kinase dimerization through steric occlusion of its domains[45,46]. Our results suggest that the 14-3-3-mediated regulation of Nedd4-2 requires simultaneous binding of two phosphorylated motifs followed by both the conformational change and steric occlusion of several functional domains, thus resembling regulations mentioned above. The CORAL model of Nedd4-2[186–975] alone indicated close interactions between the HECT and WW2/3 domains and full exposure of the WW1 and WW4 domains to the solvent (Fig. 7c and S5). This arrangement of domains corroborates previous evidence of the interaction between WW domains of Nedd4-2 and its own weak PY motive located within the C-lobe of HECT (Fig. 7c)[13,47]. On the other hand, 14-3-3 binding blocked interactions between HECT and WW2/3 by sequestering WW3 within the central channel of the 14-3-3 dimer and by weakening the interaction between the N- and C-lobes of the HECT domain (Fig. 7d and S6a). Furthermore, in a complex with 14-3-3, the WW4 domain of Nedd4-2 is located near the surface of the 14-3-3η dimer, whereas WW1 and WW2 are fully exposed to the solvent (Supplementary Fig. S6a). Such position of WW2, far from the surface of the 14-3-3η dimer, may also explain why we were unable to trace this domain in the electron density map of Nedd4-2[335–455] bound to 14-3-3η (Fig. 5e). Because WW domains presumably mediate the interaction between Nedd4-2 and its substrates, such occlusions or exposures likely affect substrate ubiquitination, accounting for the 14-3-3-mediated modulation of the ubiquitination of some Nedd4-2 substrates[36,48–50]. Considering the above, our data provide the structural glimpse into 14-3-3-mediated Nedd4-2 regulation, showing that 14-3-3 protein regulates multidomain binding partners through several common mechanisms, either promoting or blocking interdomain interactions and sterically occluding functional surfaces, among other alterations.

In conclusion, Nedd4-2 is phosphorylated on multiple sites by PKA, but dual phosphorylation on the sites Ser[342] and Ser[448] mediates high-affinity Nedd4-2 bonding to 14-3-3η. Upon binding, 14-3-3η induces a structural rearrangement of Nedd4-2

by altering interactions between the structured domains of Nedd4-2, including the N- and C-lobes of the catalytic HECT domain. Changes in the exposure of WW domains may explain how 14-3-3 modulates the ubiquitination of some Nedd4-2 substrates. For this reason, further studies should be conducted to identify the exact mechanisms of 14-3-3-dependent regulation of ubiquitination of particular substrates and to assess whether these mechanisms include variations in Nedd4-2 phosphorylation and whether the suggested conformational change of the HECT domain has any functional consequences. For now, our findings lay the foundations for future research aimed at understanding the versatile regulatory roles of 14-3-3 proteins in the regulation of signaling pathways and processes linked to protein degradation through the ubiquitin-proteasome system. Such studies are particularly relevant because many other E3 ubiquitin protein ligases, e.g., parkin, ZNRF2, CBL, ITCH and SMURF1, are also regulated in a 14-3-3-dependent manner[51–56]. Moreover, recent advances in the development of small molecule compounds targeting protein-protein interactions have demonstrated the potential to modulate the activity of key 14-3-3 binding partners in various physiological processes by targeting their distinct interactions with 14-3-3[57]. In this context, the interaction between Nedd4-2 and 14-3-3 proteins should be a promising target for the treatment of Nedd4-2-associated diseases.

## Methods

**Heterologous expression and purification of 14-3-3 protein isoforms.** All seven 14-3-3 protein isoforms (β, γ, ε, ζ, η, σ and τ) and the C-terminally truncated 14-3-3γ isoform (14-3-3γΔC, residues 1-235) were expressed in *E. coli* BL21(DE3) cells (Novagen) using a modified pET-15b plasmid with a TEV cleavage site. After affinity chromatography, the His6-tag was cleaved by TEV protease, followed by anion-exchange chromatography (HiTrap Q column; GE Healthcare)[43,58]. The final purification step was size-exclusion chromatography (HiLoad Superdex 75; GE Healthcare) in a buffer containing 20 mM Tris·HCl (pH 7.5), 150 mM NaCl, 5 mM DTT and 10% (w/v) glycerol. All isoforms were concentrated to 30 mg.ml⁻¹, frozen in liquid nitrogen and stored in aliquots in –80 °C (193.15 K).

**Heterologous expression, purification and phosphorylation of Nedd4-2.** *Nedd4-2* coding sequences (residues 335-455 and 186-975) were PCR-amplified from the plasmid hNedd4-2 (a gift from Christie Thomas, Addgene plasmid # 83433)[59]. The PCR product containing residues 335-455 was ligated into the pRSFDuet-1 (Novagen) using the *NcoI/NotI* restriction sites. Modified pRSFDuet-1 contained the sequence of the His6-tagged GB1 domain of protein G inserted into the first multiple cloning site (a gift from Evzen Boura, IOCB CAS). The PCR product, including residues 186-975, was subcloned into the expression vector pST39 (a gift from Evzen Boura, IOCB CAS) using the *XbaI/KpnI* restriction sites. The entire cloned regions were confirmed by sequencing. Mutations of *Nedd4-2* with a different number of PKA phosphorylation sites (at positions 342, 367 and 448) were generated by mutating other sites to alanine, using the QuikChange™ approach (Stratagene), and confirmed by sequencing. Oligo sequences are provided in Supplementary Table S5.

Nedd4-2 fusion proteins were expressed as fusion proteins with an N-terminal His6-GB1-tagged fusion protein (Nedd4-2[335–455] variant) or a non-cleavable 6 × His-tag at the C-terminus (Nedd4-2[186–975] variant) in *Escherichia coli* BL21 (DE3) (Novagen) cells grown in Luria-Bertani media, inducing expression by adding 0.5 mM IPTG (isopropyl β-D-1-thiogalactopyranoside) at OD600 = 0.8 for 20 h at 18 °C. The pelleted cells were suspended in lysis buffer (1 × PBS, 1 M NaCl, 4 mM β-mercaptoethanol and 2 mM imidazole) and purified using a Chelating Sepharose™ Fast Flow column (GE Healthcare), according to the standard protocol. For Nedd4-2[186–975], Tergitol NP-40 (Sigma) was added to all buffers at a final concentration of 0.01% (v/v).

The eluted Nedd4-2[335–455] protein and its variants were dialyzed against the buffer containing 20 mM Tris-HCl (pH 7.5), 2 mM EDTA, 2 mM 2-mercaptoethanol and 10% (w/v) glycerol. The His6-GB1 tag was cleaved by incubation with the TEV protease (750 U of TEV per 1 mg of fusion protein) in dialysis overnight at 4 °C. Nedd4-2[335–455] was phosphorylated by incubation at 30 °C for 2 h and then overnight at 4 °C with 1300 U of PKA (Promega) per 1 mg of protein in the presence of 0.75 mM ATP and 20 mM MgCl2. TEV and PKA were removed through another immobilized metal affinity chromatography and subsequent size-exclusion chromatography using a HiLoad™ 26/600 Superdex™ 75 pg column (GE Healthcare) in a buffer containing 20 mM Tris-HCl (pH 7.5), 150 mM NaCl, 1 mM TCEP and 10% (w/v) glycerol. The typical yield was 5 mg of pure protein per one liter of LB media.

The eluted Nedd4-2[186–975] was purified by size-exclusion chromatography directly after Ni²⁺ affinity chromatography, using a HiLoad™ 26/600 Superdex™

200 pg column (GE Healthcare) in a buffer containing 50 mM Tris-HCl (pH 8.0), 500 mM NaCl, 1 mM TCEP and 10% (w/v) glycerol and 0.01% (v/v) NP-40. Purified Nedd4-2[186–975] was phosphorylated with 250 U of PKA (Promega) per mg of protein in the presence of 0.75 mM ATP and 20 mM MgCl₂ and incubated at 30 °C for 3 h followed by size-exclusion chromatography using a Superdex™ 200 Increase 10/300 GL column (GE Healthcare) in a buffer containing 50 mM Tris-HCl (pH 8.0), 500 mM NaCl, 1 mM TCEP and 10% (w/v) glycerol. The typical yield was 2 mg of pure protein per one liter of LB media.

The level of the phosphorylation of both proteins was confirmed by the mass spectrometry service provided by CMS Biocev.

**Heterologous expression and purification of Uba1, Ube2d2 and Ub.** The coding sequences of mouse *Uba1* (ubiquitin-like modifier-activating enzyme 1), mouse *Ube2d2* (ubiquitin-conjugating enzyme E2D2) and human Ub were kindly provided by Dr. Silhan (IOCB, CAS). Both enzymes and human ubiquitin were expressed in *Escherichia coli* BL21 (DE3) cells (Novagen). The E1 enzyme Uba1 was subcloned into a pET28a plasmid with a TEV-cleavable 6 × His-tag at the N-terminus and expressed at 16 °C O/N. The E2 enzyme Ube2d2 and human ubiquitin Ub were subcloned into a pET15b plasmid with a TEV-cleavable 6 × His-tag at the N-terminus and expressed at 25 °C O/N. All these fusion proteins were expressed in Luria-Bertani media by induction at OD[600] = 0.8 for 20 h. The pelleted cells were suspended in lysis buffer (1 × PBS, 1 M NaCl, 4 mM β-mercaptoethanol and 2 mM imidazole) and purified using a Chelating Sepharose™ Fast Flow column (GE Healthcare) according to the standard protocol. The fusion proteins were dialyzed into a buffer containing 20 mM Tris·HCl (pH 7.5), 150 mM NaCl, 2 mM DTT and 10% (w/v) glycerol. The His₆-tag of Uba1 and Ube2d2 were cleaved by incubation with TEV protease at 30 °C for 2 h. The final purity of all proteins was confirmed by size-exclusion chromatography on Superdex™ 200 Increase 10/300 GL (GE Healthcare) or Superdex™ 75 Increase 10/300 GL (GE Healthcare) columns in a buffer containing 20 mM Tris·HCl (pH 7.5), 150 mM NaCl, 2 mM DTT and 10% (w/v) glycerol. The proteins were concentrated as required, aliquoted and flash-frozen in liquid nitrogen.

**Ubiquitination assay.** For in vitro ubiquitination assays, 5 μg of human Ub, 50 ng of E1 (mouse Uba1), 200 ng of E2 (mouse Ube2d2) and 500 ng of E3 (phosphorylated Nedd4-2[186–975]) were incubated in 30 μl of the reaction mixture in a buffer containing 50 mM Tris pH 7.5, 2 mM ATP, 5 mM MgCl₂ and 2 mM DTT and incubated for 5, 10 and 15 min at 30 °C. The reactions were stopped by adding 5 × SDS-PAGE loading buffer (250 mM Tris pH 6.8, 50% glycerol, 500 mM DTT and 10% SDS) and incubating at 95 °C for 3 minutes to denature the samples. For reactions with 14-3-3η, Nedd4-2[186–975] was mixed with 14-3-3η in a 1:2 molar ratio and incubated for 30 minutes on ice before starting the reaction. In addition, 20 μl of reaction was separated by conventional SDS-PAGE on a 10% acrylamide gel and electro-blotted against a PVDF membrane in 20 mM Tris pH 7.5, 154 mM glycin and 10% methanol buffer overnight at 4 °C. Rabbit anti-Ub polyclonal antibody (Enzo) and anti-rabbit IgG, HRP-linked Antibody (Cell Signaling Technology) were used to visualize the transferred polyubiquitin chains. Chemiluminescence was induced by ECL and detected using Fusion Solo S (Vilber). The detected bands of E3-Ub were quantitated using the Image Lab software (Bio-Rad).

**Fluorescence polarization assay.** The FP assay was performed using a CLAR-IOstar microplate reader (BMG Labtech, Thermo Fisher Scientific, Waltham, MA, USA) on 384-well black low-volume flat-bottom plates (Corning, New York, USA) in a buffer containing 10 mM HEPES (pH 7.4), 150 mM NaCl, 0.1% (v/v) Tween 20 and 0.1% (w/v) BSA. Seven 14-3-3 protein isoforms at a starting concentration of 80 μM, followed by binary dilution series, were incubated for 1 h with 50 nM of FITC-labelled synthetic peptides FITC-PRSLpSSPTVT (pS342), FITC-ARSSpTVTGGE (pT367) and FITC-LRSCpSVTDAV (pS448) where pS/pT denotes phosphoserine/phosphothreonine (Pepscan Presto BV). The excitation and emission wavelengths were 482 nm and 530 nm, respectively. The $K_D$ values were determined as the mean of three independent measurements using GraphPad Prism version 8.0.1 for Windows, GraphPad Software, La Jolla California USA, www.graphpad.com.

**Crystallization, data collection and structure determination.** To crystallize the complex between 14-3-3γΔC (residues 1-234) and the synthetic peptides pep-S342 or pep-S342, 16 mg.ml⁻¹ 14-3-3γΔC was mixed with the Nedd4-2 synthetic peptides pep-S342 (LRSCpSVTDAV) or pep-S342 (PRSLpSSPTVT) at a 1:1.5 molar stoichiometry, in a buffer containing 20 mM HEPES (pH 7.0), 2 mM MgCl₂ and 2 mM TCEP. Crystals were grown from drops consisting of either 2 μl of the pep-S342:14-3-3γΔC complex and 2 μl of 100 mM HEPES (pH 7.5), 200 mM MgCl₂, 23% (v/v) PEG 400, and 2% hexafluoro-2-propanol or 2 μl of the pep-S448:14-3-3γΔC complex and 2 μl of 100 mM HEPES (pH 7.5), 200 mM MgCl₂, 16% (v/v) PEG 400, and 1% hexafluoro-2-propanol, respectively. Crystals were cryoprotected using 30% (v/v) PEG 400 and flash frozen in liquid nitrogen before data collection in oscillation mode at beamline 14.1 of the BESSY synchrotron.

To crystallize Nedd4-2[335–455] in a complex with 14-3-3ηΔC, the mutant variant T367A was used to prevent sample heterogeneity. The 14-3-3ηΔC:Nedd4-2[335–455]T367A complex was mixed in a 1:2 molar stoichiometry and dialyzed overnight against 20 mM

Tris-HCl pH 7.5 and 1 mM TCEP buffer. The protein complex was concentrated to ~11 mg.ml⁻¹. Crystals were grown from drops consisting of 4 μl of the protein complex, 2 μl of Morpheus C12 condition (composed of 0.03 M Sodium nitrate, 0.03 M Sodium phosphate dibasic, 0.03 M Ammonium sulfate, 0.1 M Bicine, 12.5% v/v MPD; 12.5% PEG 1000; 12.5% w/v PEG 3350) pH 7.5 and 0.6 μl of 30% w/v sucrose (Hampton Research Additive Screen). Crystals were flash frozen in liquid nitrogen without any additional cryoprotection before data collection in oscillation mode at the D8 Venture system (Bruker, MA, USA).

Diffraction datasets were processed using the packages XDS and XDSAPP[60,61]. Crystal structures were solved by molecular replacement in MOLREP[62], using the structures of the 14-3-3η (PDB ID: 2C63), 14-3-3γΔC:pep139 (PDB ID: 6GKF) as search models and refined at a resolution of 1.8 Å (14-3-3γΔC:pepS342), 1.9 Å (14-3-3ηΔC 14-3-3γΔC:pep-S448) and 2.3 Å (14-3-3ηΔC:Nedd4-2[335–455]T367A), respectively, with the PHENIX package[34,63]. The atomic coordinates and the structure factors of the complexes have been deposited in the RCSB PDB under accession codes 6ZBT, 6ZC9 and 7NMZ. All structural figures were prepared with PyMOL (https://pymol.org/2/).

**Analytical ultracentrifugation (AUC) measurements.** Sedimentation velocity (SV) experiments were performed using a ProteomLab™ XL-I, Beckman Coulter analytical ultracentrifuge[64]. The samples were dialyzed against a buffer containing 20 mM Tris-HCl (pH 7.5), 150 mM NaCl, and 1 mM TCEP before analysis. The samples with the longer construct of Nedd4-2[186–975] were dialyzed against a buffer containing 50 mM Tris-HCl (pH 8.0), 500 mM NaCl, 1 mM TCEP and 0.01% (v/v) Tergitol NP-40 (Sigma). SV AUC experiments were conducted at various loading concentrations of 14-3-3η, Nedd4-2[186–975], Nedd4-2[355–455] and its mutant variants in charcoal-filled Epon centerpieces with a 12-mm optical path length at 20 °C, and at 40000 or 46000 rpm rotor speed (An-50 Ti rotor, Beckman Coulter). The buffer density, viscosity, and partial specific volume of all proteins were estimated using the program SEDNTERP. All sedimentation profiles were collected with absorbance optics at 280 nm. The sedimentation coefficients $c(s)$ distributions were calculated from raw data using the SEDFIT software package[65]. The calculated distributions were integrated to establish the weight-average sedimentation coefficients corrected to 20 °C and the density of water, $s_{w(20,w)}$. The $s_w$ values were plotted as a function of Nedd4-2[355–455] or 14-3-3η concentration to construct $s_w$ isotherms. The resulting isotherms were fitted with a A + B ⇌ AB model, as implemented in the SEDPHAT software package with previously known $s_w$ values of each component. The parameters were verified, and the loading concentrations were corrected using global Lamm-equation modeling, also implemented in the SEDPHAT software[66]

**Chemical Crosslinking combined with mass spectrometry.** Both Nedd4-2[186–975] alone and in a complex with 14-3-3η were crosslinked with homobifunctional crosslinkers DSG and DSS in 1:1 (mol/mol) mixtures of non-deuterated and four-times-deuterated compounds (d0/d4). For the crosslinking reaction, the proteins were dialyzed in a buffer containing 20 mM HEPES (pH 7.5), 150 mM NaCl and 1 mM TCEP. In all samples, the protein concentration was 0.6 mg.ml⁻¹. Freshly prepared stock solutions of crosslinkers (10 mg.ml⁻¹ in DMSO) were added to proteins at a 50 × molar excess in a total reaction volume of 20 μl and incubated for 1 h at room temperature.

For quantitative studies, Nedd4-2[186–975] in the presence of 14-3-3η and Nedd4-2[186–975] alone were incubated with a 50-fold molar excess of light (12 C) and heavy (13 C) disuccinimidyl adipate (DSA; Creative Molecules). After a 30-min incubation at room temperature, Nedd4-2[186–975] in the presence of 14-3-3η labelled with 13 C and Nedd4-2[186–975] alone labelled with 12 C were mixed at a 1:1 ratio.

The proteins were reduced with 10 mM dithiothreitol, alkylated with 30 mM iodoacetamide, and digested overnight with trypsin at 37 °C. The peptides were injected into a Luna Omega 5μm Polar C18 100 Å 20 × 0.3 mm column (Phenomenex) and desalted at 20 μL/min for 5 min. Peptides were then separated by reversed-phase chromatography with a Luna Omega 3μm Polar C18 100 Å 150 × 0.3 mm column (Phenomenex) at 10 μL/min using a capillary UHPLC 1290 system (Agilent Technologies) with a gradient sequence of 1−10% for 1 min, 10−45% for 19 min, and 45−95% for 5 min of buffer solvent A (0.1% formic acid, 98% acetonitrile in water) in buffer solvent B (0.1% formic acid, 2% acetonitrile in water). The column was heated to 50 °C and directly connected to a 15 T solariX XR FT-ICR mass spectrometer (Bruker Daltonics, USA) operated in positive data dependent mode. StavroX software[67] was used to identify crosslinked peptides. Fixed carbamidomethylation of cysteines and variable methionine oxidation were set as modifications. The modification sites of DSG and DSS were N-termini, lysines, serines, threonines and tyrosines. The mass error threshold was set to 1 ppm, and all assigned fragments were manually curated. For quantitative studies, the Links software[68,69] was used to identify crosslinked peptides. The nonoverlapping isotopes of peptides crosslinked with 12 C and 13 C DSA were used to calculate isotope ratios[70].

**Small-angle X-ray scattering (SAXS).** SAXS data were collected on the European Molecular Biology Laboratory (EMBL) P12 beamline, at the storage ring PETRA III (Deutsches Elektronen Synchrotron (DESY), Hamburg, Germany). The SEC-SAXS

measurements were conducted in a buffer containing 20 mM Tris-HCl (pH 7.5), 150 mM NaCl, 1 mM TCEP, 3% (w/v) glycerol using Superdex 200 Increase 5/150 GL column (GE Healthcare) at the flow rate 0.5 ml.min$^{-1}$ at 293.15 K.

The loading concentrations of the samples were 2.2 mg.ml$^{-1}$ for Nedd4-2$^{186-975}$, 2 mg.ml$^{-1}$ for 14-3-3η and 8.9 mg.ml$^{-1}$ for the Nedd4-2$^{186-975}$:14-3-3η complex (1:2 molar stoichiometry). The injection volume of the samples was 52 μl. Elution profiles with the corresponding frames were processed using ATSAS 3.0.2 software CHROMIXS[71]. ATSAS software PRIMUS and GNOM were used to determine the excluded volume of the hydrated particle ($V_p$), the radius of gyration ($R_g$) and maximum particle dimension ($D_{max}$)[72,73]. The molecular mass ($M_W$) was determined using a methods described by Rambo et al.[74]. Scattering profiles, $P(r)$ functions, normalized Kratky plots and Guinier approximations were visualized using GraphPad Prism version 8.0.1 for Windows (GraphPad Software, La Jolla California USA, www.graphpad.com). The program DAMMIF was used to calculate the *ab inicio* molecular envelopes[75]. For the complex, 10 surfaces were generated in slow mode and averaged using DAMAVER[76]. The averaged surfaces were used as the final filtered model of the complex, which was superimposed to the structural model using the SUPCOMB program (EMBL, Hamburg, Germany)[77].

**Structural modeling**. The three-dimensional models of Nedd4-2$^{186-975}$ and the Nedd4-2$^{186-975}$:14-3-3η complex were generated using CORAL. SAXS data and distance constraints derived from intramolecular crosslinks of Nedd (for model of Nedd4-2$^{186-975}$ alone), intermolecular crosslinks between 14-3-3η and the Nedd4-2$^{186-975}$ and intramolecular crosslinks of the complexed Nedd4-2$^{186-975}$ (for model of the Nedd4-2$^{186-975}$:14-3-3η complex) were used[78]. The starting model of the complex was prepared using the crystal structures of 14-3-3η (PDB ID: 2C63)[79] and 14-3-3γ with the Nedd4-2 phosphopeptides pep-S342 and pep-S448 (PDB ID: 6ZBT and 6ZC9), as well as known solution structures of the WW1-3 (PDB ID: 1WR3, 1WR4, 1WR7) and HECT (PDB ID: 5HPK) domains[80]. The unknown structure of the WW4 domain was generated using the SWISS-MODEL server (https://swissmodel.expasy.org/). Using the CORAL program, disordered loops missing in the crystal structures were modelled as interconnected dummy residue chains attached to the appropriate Cα atoms in rigid domains, but the linker length was limited to 5−100 residues. For this reason, the part of the Nedd4-2 region connecting the WW1 domain to the pSer$^{342}$ motif (residues Ser$^{233}$-Phe$^{249}$) was modeled as the α-helix based on secondary structure prediction by PSIPRED (psipred@cs.ucl.ac.uk). The best-scoring CORAL model was chosen according to the lowest $\chi^2$, and the distances were derived from intra- and inter-molecular crosslinks.

**Statistics and reproducibility**. Results from the FP assay (Fig. 4) and immunoblot (Supplementary Fig. S1) are represented as means ± SD from three replicates as indicated in the figure legend. Statistical analysis for the data from immunoblot were performed using Graph-Pad Prism 8.4. Student's *t*-test was used for comparison of relative changes of samples with and without 14-3-3η at selected timepoints (ns, non-significant $P > 0.05$; *, $P < 0.05$).

**Reporting summary**. Further information on research design is available in the Nature Research Reporting Summary linked to this article.

## Data availability

The authors declare that all data supporting the findings of this study are available within the article and its supplementary information file. Crystallography data have been deposited in the RCSB PDB with the accession codes: 6ZBT, 6ZC9 and 7NMZ. All source data underlying the graphs presented in the main and supplementary figures are made available in Supplementary Data 1.

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

## Acknowledgements

This study was funded by the Czech Science Foundation (V.O., grant number 20-00058S), the Grant Agency of Charles University (P.P. grant number 740119), the Czech Academy of Sciences (RVO:67985823 of the Institute of Physiology). We thank the Czech Infrastructure for Integrative Structural Biology (CIISB) for access to the CMS facilities at BIOCEV (project LM2015043 by MEYS) and MetaCentrum CESNET for the using of their computing clusters. We thank EMBL MX beamline 14.2 (BESSY, Berlin) and SAXS beamline P12 (Petra III DESY, Hamburg) for the allocated experimental beam time. We thank D. Kalabova, G. Kocarova and A. Smidova for technical assistance, P. Pompach and P. Vankova for their help with MS measurements and Carlos V. Melo for editing the article.

## Author contributions

V.O. and T.O. conceived the study and provided scientific guidance. P.P. and R.J. prepared the recombinant proteins. P.P. performed F.P. assays, SAXS data processing and SAXS-based modeling and crystalized the 14-3-3ηΔC:Nedd$^{335-455}$ protein complex. P.P. performed the ubiquitination assay. O.P. performed SV-AUC and analyzed data. R.J. prepared samples for chemical crosslinking. V.O., P.P. and R.J. refined the crystal structures. V.O. and T.O. wrote the paper. All co-authors revised the manuscript.

## Competing interests

The authors declare no competing interests.
