## [Peer Review File · Communications Biology]

Referee expertise:

Referee #1: expertise in 14-3-3 biology

Referee #2: expertise in ubiquitination

Reviewers' comments:

Reviewer #1 (Remarks to the Author):

In this report the authors have elucidated the structural basis of 14-3-3-mediated Nedd4-2 regulation using impressive structural and biochemical analysis. This was done as a function of phosphorylation at two sites in the Nedd4-2 molecule. This line of research has not been previously followed.

This report is convincing, novel and will be of interest to scientists in many fields because 14-3-3 is so widely distributed in biology.

It would be of interest to others if the authors were to highlight the similarities and differences in binding of 14-3-3 and Nedd4-2 (E3 ubiquitin protein ligases) compared to the binding of 14-3-3/other proteins especially as regards the roles of phosphorylation and HECT and WW domains. Is there a common mechanism for 14-3-3 binding to AANAT ? for all 14-3-3 partners?

Reviewer #2 (Remarks to the Author):

Review of COMMSBIO-21-1027

Nedd4-2, a member of Nedd4 family of ubiquitin ligases, is an important regulator of a number of membrane proteins, including several channels and transporters. In the kidney and lung epithelia, Nedd4-2 controls the membrane levels of epithelial sodium channel (ENaC) to regulate sodium homeostasis. Nedd4-2 itself is regulated by a number of hormones (e.g. insulin and aldosterone), which activate kinases like Akt and Sgk1. These kinases then phosphorylate Nedd4-2 at specific sites promoting binding of some of the 14-3-3 isoforms. 14-3-3 binding inhibits substrate binding, and presumably also the autoactivation of Nedd4-2, thus blocking its function when increased ENaC is required for sodium reabsorption. The authors have studied the 14-3-3 Nedd4-2 interactions using a number of different approaches combined with extensive structural analyses. They confirm some of the previous observations and provide additional data to demonstrate that phosphorylated Ser342 and Ser448 are the main residues that facilitate 14-3-3 protein binding to Nedd4-2. They show that upon binding, 14-3-3 induces a structural rearrangement of Nedd4-2 protein by altering interactions between the structured domains of Nedd4-2, including the N- and C-lobes of the HECT ligase domain. Their model provides new information how the exposure of WW domains to 14-3-3 could affect substrate binding and ubiquitination.

Overall I found the data compelling and interesting, and experiments well performed. The studies are a significant addition to how Nedd4 family members are regulated through binding adaptor molecules. In my opinion, the work can be published without any additional data and only with minor textual changes.

Some of the cited references require a recheck. For example, the first two references should be replaced by the more appropriate ones as follows:

1. Biochem Biophys Res Commun. 1992 Jun 30;185(3):1155-61.
doi: 10.1016/0006-291x(92)91747-e. PMID: 1378265

2. Genomics. 1997 Mar 15;40(3):435-43.
doi: 10.1006/geno.1996.4582. PMID: 9073511

Insulin signalling also leads to Nedd4-2 phosphorylation. This could be added in the introduction with an appropriate reference.

June 4th 2021

We would like to thank the reviewers for positive reviews. We have addressed all issues and changed our manuscript accordingly.

Yours sincerely,

Veronika Obsilova, PhD.

Reviewer #1:

In this report the authors have elucidated the structural basis of 14-3-3-mediated Nedd4-2 regulation using impressive structural and biochemical analysis. This was done as a function of phosphorylation at two sites in the Nedd4-2 molecule. This line of research has not been previously followed.

This report is convincing, novel and will be of interest to scientists in many fields because 14-3-3 is so widely distributed in biology.

1. It would be of interest to others if the authors were to highlight the similarities and differences in binding of 14-3-3 and Nedd4-2 (E3 ubiquitin protein ligases) compared to the binding of 14-3-3/other proteins especially as regards the roles of phosphorylation and HECT and WW domains. Is there a common mechanism for 14-3-3 binding to AANAT? for all 14-3-3 partners?

Answer: As suggested by the reviewer, we added more discussions about the similarities in binding of 14-3-3 and Nedd4-2 compared to other well-characterized 14-3-3 binding partners (page 10) and added the references # 43 and # 44 for the 14-3-3:AANAT complex.

Reviewer #2:

Nedd4-2, a member of Nedd4 family of ubiquitin ligases, is an important regulator of a number of membrane proteins, including several channels and transporters. In the kidney and lung epithelia, Nedd4-2 controls the membrane levels of epithelial sodium channel (ENaC) to regulate sodium homeostasis. Nedd4-2 itself is regulated by a number of hormones (e.g. insulin and aldosterone), which activate kinases like Akt and Sgk1. These kinases then phosphorylate Nedd4-2 at specific sites promoting binding of some of the 14-3-3 isoforms. 14-3-3 binding inhibits substrate binding, and presumably also the autoactivation of Nedd4-2, thus blocking its function when increased ENaC is required for sodium reabsorption. The authors have studied the 14-3-3 Nedd4-2 interactions using a number of different approaches combined with extensive structural analyses. They confirm some of the previous observations and provide additional data to demonstrate that phosphorylated Ser342 and Ser448 are the main residues that facilitate 14-3-3 protein binding to Nedd4-2. They show that upon binding, 14-3-3 η induces a structural rearrangement of Nedd4-2 protein by altering interactions between the structured domains of Nedd4-2, including the N- and C-lobes of the HECT ligase domain. Their model provides new information how the exposure of WW domains to 14-3-3 could affect substrate binding and ubiquitination.

Overall I found the data compelling and interesting, and experiments well performed. The studies are a significant addition to how Nedd4 family members are regulated through binding adaptor molecules. In my opinion, the work can be published without any additional data and only with minor textual changes.

1. Some of the cited references require a recheck. For example, the first two references should be replaced by the more appropriate ones as follows:

1. Biochem Biophys Res Commun. 1992 Jun 30;185(3):1155-61.
doi: 10.1016/0006-291x(92)91747-e. PMID: 1378265

2. Genomics. 1997 Mar 15;40(3):435-43.
doi: 10.1006/geno.1996.4582. PMID: 9073511

Insulin signalling also leads to Nedd4-2 phosphorylation. This could be added in the introduction with an appropriate reference.

Answer: Reference # 1 and 2 were replaced as suggested by the reviewer. Moreover, the insulin signalling with new ref. # 18 was added to the introduction (page 3).